# What influences whether parents recognise COVID-19 symptoms, request a test and self-isolate: A qualitative study

Lisa Woodland [1,2] *, Fiona Mowbray[1,2], Louise E. Smith [1,2], Rebecca K. Webster[3], Richard Amlôt[2,4], G James Rubin[1,2]

1 Department of Psychological Medicine, King's College London, London, United Kingdom, 2 NIHR Health Protection Research Unit in Emergency Preparedness and Response, London, United Kingdom, 3 Department of Psychology, University of Sheffield, Sheffield, United Kingdom, 4 Behavioural Science and Insights Unit, UK Health Security Agency, Salisbury, United Kingdom

* lisa.woodland@kcl.ac.uk

**Data Availability Statement:** Our data is deposited in the King's College London research data repository, KORDS, (https://doi.org/10.18742/17263445). The data include sensitive information

## Abstract

### Background

Using test, trace and isolate systems can help reduce the spread of COVID-19. Parents have the additional responsibility of using these systems for themselves and acting on behalf of their children to help control COVID-19. We explored factors associated with the use of England's NHS Test and Trace service among parents of school-aged children.

### Methods

One-to-one telephone interviews with parents (n = 18) of school-aged (4 to 18 years) children living in England between 30 November to 11 December 2020. Data were explored using thematic analysis.

### Results

Three themes and eight sub-themes emerged. In terms of recognising symptoms of COVID-19, parents needed prompting before recalling the main symptoms described by the NHS. Parents suggested several factors relating to the nature of the symptom(s) and contextual information that might lead to or prevent them from seeking a test. Although parents supported symptomatic testing and described trusting official sources of information (e.g., Government and NHS websites). However, some concerns were raised regarding the accuracy of test results, safety at testing centres and logistics of testing but none of the concerns appeared to prevent engagement with testing. Parents perceived adherence to testing and self-isolation as pro-social behaviour, although family resources and circumstances impacted their ability to adhere fully.

### Conclusions

Our study identified several barriers to parents using NHS Test and Trace as needed. Information about the eligibility of testing (main symptoms of COVID-19 and the age of eligibility)

and cannot be made openly available. Requests for access to relevant excerpts of the transcripts will be considered if made by academic teams willing to sign a standard access agreement and should be addressed to the King's College London research data repository team at research.data@kcl.ac.uk.

**Funding:** This study was funded by the Economic and Social Research Council [grant number ES/P000703/1] and the National Institute for Health Research Health Protection Research Unit (NIHR HPRU) [grant number NIHR200890] in Emergency Preparedness and Response, a partnership between UK Health Security Agency, King's College London and the University of East Anglia. The views expressed are those of the authors and not necessarily those of the NIHR, Public Health England or the Department of Health and Social Care.

**Competing interests:** GJR, RA and LS participate in the UK's Scientific Advisory Group for Emergencies, or its subgroups. These groups did not fund the study or authors. RA is an employee of the UK Health Security Agency. The author interests do not alter our adherence to PLOS ONE policies on sharing data and materials.

**Abbreviations:** NHS, National Health Service; NHSTT, NHS Test and Trace; COVID-19, Coronavirus disease; PCR, Polymerase chain reaction (test; LFT, Lateral Flow Test.

needs to be more precise and resources provided to enable families to adhere to self-isolation if the efficiency of test, trace and isolate systems is to be optimised.

## Introduction

NHS Test and Trace (NHSTT) was launched across England in May 2020, helping to identify, contain and control the spread of SARS-CoV-2 [1]. NHSTT advisers get in touch with anyone who tests positive for COVID-19 (cases) and notify people they have had recent interactions with (contacts). By law, cases must self-isolate for ten days and contacts for fourteen days (reduced to ten days from 14 December 2020, and more recently limited to people who are unvaccinated) [2]. People in self-isolation must stay at home and are not permitted to leave the home for any reason, except for a COVID-19 test.

The success of NHSTT relies on members of the public requesting a test as soon as they experience one of the 'main' symptoms of COVID-19 listed by the NHS. At the time of this study, the 'main' symptoms emphasised to the public in England were a high temperature, a new, continuous cough, and a loss or change to your sense of smell or taste. People who experience these symptoms should self-isolate unless they receive a negative result from a polymerase chain reaction (PCR) test. Everyone in the household of a case is considered a contact and therefore must also self-isolate. A large UK study conducted between March 2020 and January 2021 found that only 51.5% of participants could identify the 'main' symptoms of COVID-19, and only 42.5% of people fully adhered to guidance around self-isolation [3]. Having a dependent child in the household was strongly associated with non-adherence [3]. The most common reason for not requesting a test when symptomatic was assuming that COVID-19 was not the cause of the symptoms.

In April 2020, Hodson, Woodland [4] interviewed 30 parents in England and found they commonly considered other, more mundane causes for their child's symptoms rather than COVID-19 as a potential cause. Parents were more likely to seek a non-COVID-19 interpretation if their child had a cough than if they had a fever or experienced unusual or unexpected symptoms. However, parents may also be anxious about their child catching COVID-19 [5], possibly increasing their attention to and the likelihood of recognising symptoms of COVID-19 in their children [6].

From September 2020, schools were required to implement "bubbles"–distinct groups of pupils which could be the size of a class or year group. The bubbles mitigate children mixing with numerous children, limiting the number of contacts needing to self-isolate when a case is identified. Between August and October 2020, hundreds of school outbreaks occurred and a period of nationwide school closures happened in the winter of 2020–21 [7]. The school closures implemented to control COVID-19 have severely affected children with negative impacts on their education and mental and physical health [8,9]. Parents, therefore, face the competing demands of minimising further disruption to their child's education and health while needing to ensure their child self-isolates when appropriate to reduce the spread of infection. These difficulties highlight the importance of identifying areas within the NHSTT process to increase parents' engagement.

In this study, we investigated perceptions and experiences relating to the use of NHSTT among parents of school-aged children (4 to 18 years) primarily to understand factors associated with COVID-19 symptom identification and the reasons why parents do or do not request a test when their child is symptomatic. We also explored the reasons for non-

adherence to self-isolation among parents and their school-aged children when someone in their household is symptomatic.

## Method

### Design

We conducted one-to-one qualitative interviews with parents of school-aged children.

### Participants

We used a specialist market research service, Angelfish Fieldwork [10], to recruit participants from their online opt-in research panel. The study was advertised alongside another study we were simultaneously running [11] between 24 November and 3 December 2020, and 1,447 potential participants expressed interest in both and were screened for eligibility. 357 potential participants were ineligible, and 771 were eligible for the other study [11], where potential participants included the general population and university students. For the current study, 319 potential participants met the eligibility criteria of being over 18 years, a parent to at least one school-aged child (4 to 18 years) and living in England. The recruitment company purposively selected a range of eligible participants for interview based on their age, gender, living region, ethnicity, marital status, whether someone in the household had experienced COVID-19-like symptoms in the last seven days and child's age, to ensure a diverse sample.

### Interview

We used a semi-structured interview guide incorporating open-ended questions asking participants to describe their perceptions and experiences of the topic. We asked parents: to describe common cold, influenza and COVID-19 symptoms and any recent experience relating to those symptoms; how they would behave if they or their child(ren) had symptoms thought to be caused by COVID-19 and whether their behaviour would differ depending on who in the household was symptomatic; what factors might prompt or deter them from getting a test, how they would get a test and whether and how they would self-isolate. Parents were also asked about their perceptions and concerns about testing. The complete interview outline is provided as supporting information S1 Text.

### Procedure

Participants referred to us by the recruitment agency were given an information sheet and asked to provide informed consent, following which an appointment was made for a telephone or online interview. Consent was also taken verbally before each interview. Two female researchers (LW and FM) conducted the interviews, both of whom had previous qualitative research experience. Interviews took place between 30 November—11 December 2020, were audio-recorded and had a mean length of 44 minutes. Participants were paid a £40 e-gift card for their time.

### Data analysis

Interviews were transcribed verbatim by an external transcription company. LW analysed data using thematic analysis, using Nvivo version 12 [12]. Key statements were grouped by topic aligning with the interview guide. Themes were extracted by topic by LW using the six-phase approach recommended by Braun and Clarke [13]. Themes were reviewed by authors at each phase of the process. We resolved disagreements through discussion until an agreement was

reached between all authors. An inductive approach was used from a positivist epistemological position. Consolidated criteria for reporting qualitative research (COREQ) were followed [14].

## Sample size

Dependent on the number of eligible participants available for interview within the interview period, we aimed to interview between 14 and 20 participants. The target sample size was agreed using Fugard and Potts [15] framework to provide a high likelihood of identifying the most prevalent themes and reach data saturation.

## Ethics

The research was approved by the Psychiatry, Nursing and Midwifery Research Ethics Sub-committee at King's College London (LRS-20/21-21336:COVID-19).

## Results

### Participants

Eighteen participants were included in the study. Participants came from all regions of England, were evenly split by parent gender and had a mean age of 49 years. All participants had at least one school-aged child (4 to 18 years), but because of multiple children living within a household, children's ages ranged from 1 to 22 years (total children, n = 38, mean age 13 years). Further demographic information is presented in Table 1.

### Theme 1: Factors affecting parents seeking a symptomatic COVID-19 test

The supporting quotes for theme 1 are shown in Table 2.

**Sub-theme 1: Recognition of COVID-19.** The main NHS listed symptoms, a high temperature, a new, continuous cough, or a loss or change to your sense of smell or taste, were often reported to be the "main," "typical," and "key" symptoms of COVD-19. However, the interviewer commonly needed to prompt parents before all three symptoms were reported.

Parents also mentioned other symptoms, particularly lethargy, tiredness, headaches, and breathing difficulties. Linked to this was the perception that COVID-19 has "lots of different symptoms" because it "affects people differently." Similarly, one parent suggested the presentation of COVID-19 symptoms in children was "very wide and varied." In response to this issue, some parents supported the Government limiting the case definition of COVID-19 to three 'main' symptoms to prevent parents needing to request a test "all the time" and mitigating anxiety about recognising COVID-19 in their children, triggered by "every single thing [the media] mention."

Parents reported monitoring their children for signs of illness, ensuring they are eating healthy and sleeping well more than they usually would and regularly asking them if they were experiencing any main symptoms of COVID-19. In addition, parents presumed that schools would also be "looking for" those symptoms and would send children home if they developed symptoms at school. Parents were mainly trusting their children to inform them when they were unwell, which was linked to the confidence they had in understanding their children's ability to describe symptoms of illness and recognise and report symptoms of COVID-19:

*"Yes, and my children can both. . . My three-year-old can talk properly and everything, so she could tell me if there was something wrong, and she probably would, so I'm not particularly concerned." (P015)*

**Table 1. Participant demographic information.**

| Demographic Information | | Frequency |
|---|---|---|
| Parent gender | Female | 9 |
| | Male | 9 |
| Parent age | 35–40 | 3 |
| | 41–45 | 2 |
| | 46–50 | 6 |
| | 51–55 | 4 |
| | 56–60 | 3 |
| Region of England | West Midlands | 4 |
| | North West | 4 |
| | London | 2 |
| | South East | 2 |
| | East Midlands | 2 |
| | Yorkshire and the Humber | 2 |
| | East of England | 1 |
| | South West | 1 |
| Ethnicity of parent | White British | 10 |
| | Pakistani | 3 |
| | Black British | 2 |
| | Indian | 2 |
| | British Indian | 1 |
| Parent employment status | Full-time employee | 13 |
| | Homemaker | 2 |
| | Self-employed | 2 |
| | Semi-retired | 1 |
| Experienced COVID-19-like symptoms in the last seven days | Parents | 6 |
| | Children | 2 |
| Number of children in the household | 1 | 4 |
| | 2 | 10 |
| | 3 | 3 |
| | 4 | 0 |
| | 5 | 1 |
| Age of children | 1–5 | 4 |
| | 6–10 | 11 |
| | 11–15 | 15 |
| | 16–20 | 7 |
| | 21–25 | 1 |

Parents were less confident in their ability to monitor a loss or change to sense of smell or taste, given that there was no observable indicator for them to use when looking for it in their children. Therefore, some parents felt they would have to rely on their child to recognise and report that symptom.

Parents used the media as a prominent source of information about the symptoms of COVID-19. In addition, parents frequently seemed to base their understanding of the symptoms of COVID-19 on their own, their children's, friends', and families' experiences of COVID-19:

**Table 2. Supporting quotes for theme 1: Factors affecting parents seeking a symptomatic COVID-19 test.**

| Sub-theme | Description | Supporting Quote |
|---|---|---|
| 1) Recognition of COVID-19 | The main symptoms of COVID-19 are a high temperature, a new, continuous cough, or a loss or change to your sense of smell or taste. | "I know that the key symptoms are a continuous cough and then high temperature and the loss of taste and smell." (P10) |
| | The other symptoms of COVID-19 are lethargy, tiredness, headaches, and breathing difficulties | "I was starting to get a little bit worried because obviously one of the main symptoms is that you have a high fever. Then there's the whole tiredness of thing could be related to a loss of breath because you're not feeling your fullest in terms of being able to stay awake and you're losing the ability to just be your normal self in terms of being able to breathe properly." (P06) |
| | Limit symptoms of COVID-19 to the three 'main' symptoms to prevent parents over-testing and mitigate anxiety about recognising COVID-19. | "Other people that I've spoken to that had it, also got sickness. You know what, if you start putting every single symptom down, then people are just going to go testing themselves all the time—there has to be main ones, doesn't there? But, for me, it didn't seem to be the three typical ones, other than, obviously, taste and smell." (P11) |
| | Parents use the media as a source of information about the symptoms of COVID-19. | "It was an article I think on Sky News and they were interviewing somebody that had had it. This was when it was first identified as a symptom and the lady in question said something that was really interesting. She said, 'I realised I'd lost my taste when I was brushing my teeth and couldn't taste the toothpaste.' So based upon that, that was an incredibly good description of how it was. So in other words, you have lost your taste, you cannot taste anything, or can't discriminate between things because something like a mint toothpaste is a very strong taste and if you can't taste that, it's bad. That actually brought it home to me." (P09) |
| | Parents trust they will be able to recognise COVID-19 symptoms in their children. | "For me, I would recognise the main symptoms—obviously, L* is only one, so he couldn't tell me—he can't talk. He couldn't tell me, so I'd be looking for a cough, in a baby who is one—and a high temperature. I wouldn't know if he's lost his taste and smell; I just wouldn't know. They were the two things that I was looking for. Again, sickness bug wouldn't even register. Obviously, G*, he's 11; he'd be able to tell me if he lost his taste and smell—but I'd be looking for the main symptoms." (P11) |
| 2) Factors impacting COVID-19 attribution | Combination of three factors relating to the symptom(s); 1) The symptom(s) present 2) The number of symptoms 3) The length of time symptoms are present | "It was the temperature and the cough that presented in my daughter and six in the morning when she woke up with a temperature, and I thought well I better get a test, I better not let this go, I'm trying to think, yeah it was, it was the end of September because I think she'd only been back to school a couple of weeks, but yeah, it was basically, she's got two out of the three symptoms so I better get her tested." (P04) |
| | Parents use contextual information to eliminate common cold and influenza as the cause. | "One of the things you do, well, that they say is that you don't necessarily get a cold with Covid. Therefore, they've had, so if they'd had a sniffle, well, that's obviously not Covid. 'There's some, whatever is, Lemsip on the side. Have of one of those and hopefully tomorrow you'll be feeling better." (P16) |
| | Specific circumstances increase parents' attributing symptoms to COVID-19. | "If they displayed symptoms, or if I felt that their cough wasn't just a cold cough. It would be my judgement, but they had other symptoms with it as well, and they'd been in touch with someone else who has tested positive. I don't want to waste anyone's time. As a mum, you know a cold is a cold. We're not paranoid. If I found out that one of their friends has had it and then they displayed cold like symptoms that could be Covid, they might not have the whole range, then I would arrange tests." (P08) |
| | Parents consider their children's personality before requesting a test | "What I would say is difficult is we have a son who suffers regularly from coughs, so that can be—when he starts coughing, you start to think is it a symptom, and then his cough disappears again. . . It's a case of some acknowledgement if it's his normal cough." (P18) |
| | Parents seek a second opinion in a medical professional after recognising symptoms | "I think doctors first because I'd probably panic, especially if it was the kids, so I think it would be doctors first, and obviously contact them. Depending on if it was over the weekend and obviously, I knew that they couldn't necessarily be contacted, then I would just go to gov.uk or just go online and search for it there and yes, take it from there really, so doctors or online to get some advice." (P03) |

"*Temperature, a persistent cough, smell, loss of taste or smell, and then I actually know some-body's who's had it, and they were just saying a really bad head and fatigue.*" *(P17)*

**Sub-theme 2: Factors impacting attribution of symptoms to COVID-19.** A combination of three factors relating to symptom(s) appeared to make it more likely that they would be attributed to COVID-19: presence of the main COVID-19 symptoms, a greater number of symptoms, and longer duration of symptoms. Parents reported intending to seek a test for the main symptoms described by the NHS; this was particularly true for a loss or change to your sense of smell or taste. Parents perceived a loss or change to your sense of smell or taste as "very specific" to COVID-19. Those who had experienced the symptom suggested it was "unusual" and "something I have never experienced," indicating the cause of the symptom as COVID-19. Parents were also more intent to seek a test if more symptoms were present:

"*I think that the symptoms you get for Covid are set. If you tick, I think more than three or four I think, then, I would definitely get the test done.*" *(P01)*

Parents were also more inclined to seek a test with increasing duration of symptoms. Usually, this occurred because parents initially adopted a "wait and see" approach to symptoms:

"*we left it [a cough] a bit of time to see if it was just one of those things. Then obviously, it wasn't going away, so we, just did the right thing [got a test].*" *(P16)*

Parents struggled to distinguish between influenza or common cold symptoms and COVID-19 but suggested that symptoms of COVID-19 last longer. Symptoms that were seen as "just a cold" rather than COVID-19 generally included runny nose, sniffles, and sneezing, although these symptoms did not necessarily deter a parent from intending to seek a test. Parents would also try to use contextual information to eliminate common cold and influenza as the cause before requesting a test. For example, symptoms that were still apparent after medication for common colds or influenza had been taken were more likely to be attributed to COVID-19, as were symptoms in somebody who had already received an influenza vaccination. Some parents reported they were more cautious because of being in a pandemic and suggested COVID-19 could have caused the symptoms. This doubt led some parents to perceive that it was "better to be safe than sorry," and therefore, they would seek a test.

Specific circumstances could also increase the likelihood of symptoms being attributed to COVID-19 because "it can't be a coincidence," particularly after potential exposure. Such events included: (1) someone in the household having been in physical contact with someone who tested positive for COVID-19, (2) someone in the household having not fully adhered to COVID-19 guidance, (3) multiple people in the household having symptoms (similar or different), within a close period.

Some parents described a need to consider their children's personality before requesting a test. Parents were less likely to request a test for a child when the parent perceived them to be prone to "exaggerate" and "imagine" symptoms of illness:

"*Yes, but unfortunately my son's got autism. He was imagining, how could I put it? He was imagining he had things wrong with him. Of course, because I had it [COVID-19], he thought he's got it, but I knew he didn't.*" *(P05)*

Parents were more likely to seek a test for a child they perceived to be rarely ill. Existing conditions that required parents to distinguish between chronic and new symptoms made decision making difficult.

Attributing symptoms in children to COVID-19 was often a "judgment call." Second opinions were sometimes sought, sometimes from a medical professional, before requesting a test. The advice received was a strong indicator of whether parents would seek a test, such as one parent who did not seek a test for his son after calling the NHS-111 helpline in September 2020:

"*Well, it was just mentioned that the, what you need to do is self-isolate. There was nothing mentioned about testing. On the hindsight, perhaps I should have said something along the lines of, 'I thought we had tests now as well,' but I didn't, so well, we didn't, we don't know, but I'm pretty convinced he had it because it was a very, very bad cough.*" (P16)

### Theme 2: Parents' perceptions and experiences of COVID-19 testing

The supporting quotes for theme 2 are shown in Table 3.

**Sub-theme 3: Eligibility for testing.**   Parents routinely reported that anyone displaying a main NHS symptom of COVID-19 was eligible for a COVID-19 test. However, parents also commonly reported other situations that made someone eligible or ineligible for testing, including referring to the elderly and people with "underlying health conditions" as eligible and young children as ineligible ("children below a certain age wouldn't," "all. . .except maybe babies"). Parents also suggested circumstances when testing is mandatory, such as a job requirement and before an operation. Furthermore, parents indicated they may seek a test to check they did not have COVID-19 before meeting others and after being exposed to someone with COVID-19:

"*Even if I would have thought there was going to be instances where you may, let's say, want to go and visit somebody in a care home, and you have a test maybe to check you're all right to visit, because, obviously, the problem with Covid is that you may have symptoms, you may have Covid, but you don't have symptoms.*" (P16)

**Sub-theme 4: Concerns about testing.**   Parents commonly indicated they had "no concerns" about COVID-19 testing for themselves or their family. Indeed, their responses reflected strong support for testing such as "yes, fine, absolutely brilliant" and "definitely."

Parents referred to "trusting" the Government, felt that providing personal information during the testing process was necessary and that the data collected would be treated professionally. Rare concerns included questions over how the Government might use DNA on the swabs later (something the participant acknowledged might be a "conspiracy theory") and "alarm bells" about the swabs having "something on them" that produces a positive result.

Some parents were unsure about the test results' accuracy and felt that self-administering the test decreased accuracy. The potential need for parents to test their children and the swabbing being "uncomfortable" exacerbated parents' concerns regarding administering the test.

A third area that raised some concerns was testing logistics. For some, unease was felt at breaking self-isolation and potentially spreading the virus when needing to return home test kits in the post, especially for single parents who had no option but to take their children with them. However, there was also the worry of "actually catching it" at testing sites and putting the family in a "vulnerable position" by being there. There were also concerns about the availability of test appointments, access to a test site and the length of time before receiving test results.

**Table 3. Supporting quotes for theme 2: Parents' perceptions and experience of COVID-19 testing.**

| Sub-theme | Description | Supporting Quote |
|---|---|---|
| 3) Eligibility for testing | Anyone displaying one of the three NHS listed symptoms are eligible for a COVID-19 test. | "Anybody who's displaying one of those three typical symptoms." (P11) |
| | Other eligibility situations, including specific groups of people, in mandatory situations and to check they did not have COVID-19. | "I think mainly elderly and people with chronic diseases." (P01) |
| | | "It's healthcare workers, it's people that work at school, it's people in care homes." (P09) |
| 4) Concerns about testing | No concerns about testing; trusting the Government is necessary. | "No, I don't think so. I did see on the—when you put all your details in—there is a data privacy statement: how they handle your data. People, if they want to, can read—I didn't—but people, if they wanted to, could. I suppose there is an element of trust with the government, so it didn't give me cause for concern: I saw that they followed all the guidelines and there was additional information, should you want to read it." (P11) |
| | Concerns relating to the accuracy of the results. | "The only concern is that if you're doing it yourself as to how accurate it is if you don't do it properly." (P18) |
| | Concerns about testing logistics. | "I would probably prefer to do a home test because, like I say, I think for me it would be a risk if I was to visit a test centre because there would be other people there who potentially have other symptoms and I could be exposed to them if I don't have Covid, and then. . . Yes, I'm just putting myself in a vulnerable position, I think if I go to a test centre." (P10) |
| 5) Information needed about testing | Parents could find any information they needed by searching the internet. | "No, I think again it goes back to one of the other things that was said there. Me personally, I'm quite comfortable about the testing, so no. If I don't know something I'll go and look it up on the government website, that is fairly comprehensive, or NHS UK or gov UK or wherever it happens to be. So I'm confident at the moment if there was any gaps and I absolutely had the burning desire to know something, I'm fairly sure that with a bit of digging I could probably find it out." (P09) |
| 6) Difficulties whilst waiting for test results | Waiting for test results appeared to be a stressful experience, worries over health and uncertainties surrounding the waiting period. | "Again, how it affected them [children]—it depends. If it affected me and I ended up in hospital, they would have anxiety over it. You worry, don't you? It's possibly death. . . The first time I had it done I was really anxious. I was really worried. You can't get it off your mind. I kept thinking, God, have I got it? I thought what's going to happen now?" (P01) |
| | Parents opposed whether to inform their children if they take a COVID-19 test. | "I suppose you wouldn't want them to worry, so maybe I wouldn't. You wouldn't tell immediate—I don't know, actually. Maybe I wouldn't tell immediate family because you wouldn't want to put that worry on them in the sense that this could develop into something serious. . .There's all this worry that you wouldn't want to put on both of your children in the sense that it's your children, they'll have seen other parents that might have sadly passed away, and now starting to think, oh hang on, that could happen to my parents.' It's just a lot of worry and concern, that I probably wouldn't put on." (P06) |
| | Reduce their anxieties by focusing on the altruistic nature of testing as a way to protect others. | "Then, obviously, there's anybody that my children have been in contact with who are vulnerable, who have low immunity, whatever, asthma perhaps, I'd be concerned for them. . . I would inform everybody I'd been in contact with and I'd be informing the schools as well from a children point of view and speak to my husband. It's informing people and workwise, I'm working from home, so that doesn't make a difference." (P10) |

**Sub-theme 5: Information needed about testing.** Parents generally assumed they could find any information they needed by searching the internet. Parents who had been to testing sites mentioned appreciating the individual advice they received from talking to someone who worked there. Using a reliable source was vital to parents, with several commenting they were only using information from the NHS, Gov.UK websites, NHS-111 telephone helpline, and, to a lesser extent, the BBC. Even parents who indicated a lack of trust in the Government's response to COVID-19 would use those sources for information. Parents also turned to friends, family and GPs for information about testing and some were concerned about their children accessing information from unreliable sources:

*"I think I always have done; always have done. I think particularly just reliability. I know speaking to my girls, it's lived by TikTok, I think now. I think they wouldn't necessarily look to the same websites and media as I do, which is very worrying." (P12)*

**Sub-theme 6: Difficulties whilst waiting for test results.** Parents with experience of testing were often pleasantly surprised with the short length of time in which they received their test results, within up to 48 hours. The testing period appeared to be a stressful experience, mainly whilst waiting for test results, with stress increasing the longer parents waited for the results. A key stressor related to worrying about the potential health implications caused by COVID-19. Uncertainties about when the test result would be received caused difficulties for parents in making plans, such as whether they needed to order food deliveries, cancel social activities, organise taking time off work and keeping their child off school.

Parents appeared to reduce their anxieties about waiting for test results by focusing on the altruistic nature of testing as a way to protect others. This made it easier to inform other people that someone in the home was awaiting a test result. Thoughts about stigma towards COVID-19 were occasionally raised, sometimes to dismiss it ("it's not like going back to the mid– 80's when AIDS was starting to come out") but sometimes because "people's reactions have been very odd." This included reports of children being bullied. As one parent noted, "COVID has definitely made people feel ostracised just from the mere mention of it, let alone having it." Parents held differing opinions on whether to inform their children if the parent had taken a COVID-19 test. For some, parents would avoid disclosing it to the family unless they needed to, feeling anxious about worrying them. Others thought they would be unable or would not want to hide it. Therefore, they would inform their children, answer questions their children may have honestly, and provide reassurance.

## Theme 3: Adhering to the guidance

The supporting quotes for theme 3 are shown in Table 4.

**Sub-theme 7: Factors impacting adherence to taking a COVID-19 test and self-isolating.** A main symptom of COVID-19 was commonly reported as a reason for requesting a test, while waiting for results and receiving a positive result were commonly a reason for self-isolating. Considering the impact of isolating on family and employers before requesting a test was also a factor. The impact of self-isolation was especially pertinent for parents who had to isolate multiple times because of their children:

*"Yes, because, I mean, a couple of times I've had to have time off work because someone in my daughter's class got tested positive and then again, when I had to do a test. About four months ago I had to take time off work, so I'm starting to feel a bit guilty and bad for my employer, so I'm definitely starting to think about it [getting a test] a lot more." (P15)*

Seven parents were particularly concerned for their child's education and missing more school after the school closures, exacerbated by children frequently needing to isolate themselves due to being a contact and reporting their children were "in and out" of school:

*"It would impact them [children] more. I'd be concerned about my children more. For the children, they'd have to stay at home from school and more missed school time, potentially becoming ill." (P10)*

However, worries about the impacts of self-isolation were often surpassed by parents intending to get a test for their children when needed, suggesting "health comes first."

**Table 4. Supporting quotes for theme 3: Adhering to the guidance on testing.**

| Sub-theme | Description | Supporting Quote |
|---|---|---|
| 7) Factors impacting adhering to taking a COVID-19 test and self-isolating | The impact on family and employers from isolating. | "I'm not sure. If anything, it makes me want to definitely double check that they do have a temperature before I even suggest that we get tested because we have to wait for the test results. I couldn't go into work for two days until I got my test results back when I had my test, so that was not particularly great. It meant that I couldn't drop my kids off at school. My husband had to leave for work late because he had to do that because I wasn't allowed to leave the house. Yes, I definitely would double check that there's definitely something wrong with them before I think about making it official and getting tested and telling playgroup or nursery." (P15) |
| | Parents requesting tests and self-isolating inappropriately. | "I think the main thing would be having the symptoms, but I think if other people in my household have caught the virus, then I'd probably think, well, it's probably worth me doing the test, just to be sure that I don't have it. Sometimes you might not have, I think it might be possible they don't show any symptoms, but you could still catch the virus. I think that's possible." (P06) |
| | Family circumstances and resources. | "Because we haven't been out, we haven't gone anywhere else, we didn't do any interactions with anybody, it's very difficult but, when we told people, we got a lot of help. We are very lucky. We've got an amazing support system in our life. We've got friends. We've got families. Obviously, people wanted to help us out; 'We'll bring some food for you guys.' "(P02) |
| | Preparing for self-isolating. | "We just didn't leave the house apart from going to the post box and I've been quite flabbergasted at the amount of people who have similar scenarios and then sort of had to cry for help and ask people to go get them a few days' worth of groceries, and I'm not judging people but at the same token I was aware that this could happen at any moment in time, more in actual fact during the winter time we had snow up here in the North of England were we got cut off for a few days, I've had flu before so I've always got lots of soup, things in the freezer, its kind of an eventuality so I didn't even have to ask somebody to go shopping for me, I just said right were on lockdown until the test comes back so we had five or six days in the house were we just stayed at home." (P04) |
| | Bombarded with calls from the NHSTT advisers. | "Great, but my feedback to you would be, it was a bit too much. When someone's not that well and you're phoning every day, sometimes you're getting your phone call two or three times a day. Then, next day, another two calls. Then, another call. You're saying the similar thing, by the way." (P02) |
| | Accessibility of tests was a pivotal resource to getting tested. | "It was a bit harder to get an appointment. I had to keep checking every hour or so for about. . . I probably had to check about 18 times before they came up with a space. They kept on recommending that I go to London to do it, which is two hours away. Then finally we found. . . We got a space near us." (P15) |
| 8) Frustration towards non-adherence | Dangerous to be spreading the virus. | "That's dangerous, absolutely. I think it's just not fair on that person and it's not fair on all the other people, right? You're just literally spreading the virus, because it's very contagious. For that reason, if you test positive, simple, it's not complicated, just lock yourself in for 14 days. That's it." (P02) |
| | Anger was apparent towards celebrities, and public figures. | "Yes, and I don't think it helps when you have celebrities who should know better, who go and flaunt the rules. That that certainly doesn't help and yes. I think that's probably one of the worst things that's come out of this, is that there's certain individuals within the world of celebrity who think they can do what they like." (P16) |

The personal responsibility to protect themselves, others and to be safe led to some parents reporting they may request a test before leaving self-isolation and after being in contact with someone who tested positive, both of which are not recommended. Another parent intended to self-isolate without taking a test, to be "cautious" after being in contact with someone with COVID-19 rather than putting others at risk by going to a testing site. Guidance at the time suggested trying to isolate the infectious person from others within the household. Parents found this difficult with younger children who still need looking after. Others would not separate people within the household, presuming everyone in the home would have COVID-19.

Family circumstances and resources impacted the ability to adhere to self-isolation. For example, some parents indicated self-isolating would have no impact on them, suggesting the

experience was no different or similar to the Government restrictions in force at the time. For others, receiving support, such as someone going shopping for them was an essential resource required for full adherence to self-isolation. Receiving social support was complex for families who routinely supported others with shopping and therefore needed to organise their shopping and those they supported. Parents needed flexibility from their employers, such as working from home and changing working hours whilst self-isolating.

Some parents prepared for self-isolation by keeping the house stocked with food, medication and booking regular food deliveries because of the unpredictability of when they may need to isolate. For parents who did not report being prepared, food shopping was a common reason for breaking isolation. Parents appeared more relaxed with continuing errands they deemed essential such as dropping the children off at school and food shopping whilst waiting for test results:

> "*The time between being ill and waiting for that test result to come back, I did go out and about. I did limit it, because I was thinking, I'm waiting for a test result—but there were some certain things that I had to do and, unfortunately, as a mum, you still need to go to the shops and you still need to do things. As soon as I got that positive result, I didn't leave the house for ten days; and then the rest of the family had to self-isolate for 14 days.*" (P11)

Parents felt "bombarded" with calls from the NHSTT advisers after receiving a positive result, which led to some feeling they wanted to lie and not list all their contacts to end the conversations knowing they had already provided this information. People who felt unwell found answering (numerous) calls particularly difficult. Trust in the system was reduced for parents who were not contacted by NHSTT when they expected to be, reducing their future engagement with NHSTT.

Accessibility of tests was a pivotal facilitator to getting tested. Testing sites being near and slots available encouraged testing. This was illustrated by one parent who "didn't have the energy" to drive to a testing site but found it "fantastic" that they could use a postal test.

**Sub-theme 8: Frustration towards non-adherence.**   Parents felt frustration towards people who did not adhere to COVID-19 rules. Parents suggested those who did not self-isolate and get a test when appropriate were "selfish" because it was "dangerous" to be spreading the virus. Parents would consider reporting people to the police, "having a word," and distancing themselves from someone they perceived as non-adherent.

Anger was apparent towards celebrities and public figures for non-adherence, such as a parent mentioning their children were "refusing to listen" to the music of a music artist in protest. Adherence was reported as not being "policed," resulting in feeling the test and trace system was "useless" and "pointless."

## Discussion

Parents face unique challenges in the context of test, trace and isolate, having to identify symptoms of COVID-19 in themselves and their child, knowing when to request a test, and potentially placing the entire household into self-isolation as soon as symptoms present. Our study identified three themes and eight sub-themes (summarised in Table 5) that help to explain why some parents do not fully adhere to COVID-19 testing and self-isolation.

Most parents were able to recall some of the main symptoms of COVID-19, indicated by NHS guidance as requiring a test, but often only listed all three main symptoms after prompting [3]. Parents also mentioned other symptoms (e.g., lethargy, tiredness, headaches, and breathing difficulties) that they would use as an indicator of COVID-19. Symptoms parents

**Table 5. Summary of the findings (three themes and eight sub-themes).**

| Theme 1:<br>Factors affecting parents seeking a symptomatic COVID-19 test | Theme 2:<br>Parents' perceptions and experiences of COVID-19 testing | Theme 3:<br>Adhering to the guidance |
|---|---|---|
| **Recognition of COVID-19**<br>–Parents trust they will be able to recognise COVID-19 in their children.<br>–Parents identify COVID-19 by specific symptoms, which is often learned via the media and from personal experience.<br>**Factors impacting COVID-19 attribution**<br>–Attribution depends on the symptom(s) present, the number of symptoms and the length the symptoms are present.<br>–Parents try to eliminate alternative causes before attributing the symptoms to COVID-19 and specific circumstances can increase parents attributing the symptom(s) to COVID-19. | **Eligibility for testing**<br>–Parents perception of eligibility included anyone with symptoms of COVID-19 and specific groups of people.<br>**Concerns about testing**<br>–Parents reported they needed to trust the Government.<br>–Some concerns about the accuracy of test results and testing logistics were raised.<br>**Information needed about testing**<br>–Information about testing was found by searching the internet.<br>**Difficulties whilst waiting for test results**<br>–Waiting for test results was found to be a stressful experience and anxieties could be alleviated by focusing on the altruistic nature of testing. | **Factors impacting adhering to taking a COVID-19 test and self-isolating**<br>–Accessibility of tests, family circumstances, resources, and the impact on family and employers were considered before engaging with testing and self-isolating.<br>**Frustration towards non-adherence**<br>–Parents perception that it was dangerous to spread COVID-19 and their anger towards others for non-adherence, increased adherence. |

identified as indicators of COVID-19 did not differ between themselves and their children. However, parents reported increased monitoring of signs of illness in their children.

There have been calls for guidance about the symptoms of COVID-19 to be broadened [16]. Our findings suggest that many parents are already adopting a relatively loose 'lay case definition.' In line with other research, we found that thinking that COVID-19 potentially caused their child's symptoms was key to parents seeking a test, but that many parents started with an assumption that symptoms were probably due to influenza or a common cold [4]. Broadening the guidance may reduce the need for parents to differentiate between the cause of symptoms and encourage more parents to seek a test for ambiguous symptoms.

On the other hand, broadening the guidance may have some adverse effects. First, we found some parents approved of limiting the symptoms to three specific symptoms because they were concerned about constantly testing themselves and their children, and reduced their anxiety that any symptom might be caused by COVID-19. Second, parents often needed prompting before recalling three symptoms. Increasing the number still further may result in greater confusion.

Our results were also in line with previous findings by Hodson, Woodland [4], that unusual symptoms were more of a concern to parents than cough and that parents were more likely to seek a test when multiple symptoms were experienced, and if symptoms persisted. A loss or change to your sense of smell or taste notably appeared to elicit parents seeking a test, with parents highlighting the symptom as uncommon.

Furthermore, parents were more likely to attribute symptoms to COVID-19 following events that could have potentially exposed the symptomatic person to COVID-19. Salience was placed on the symptoms when they occurred within a period they associated with the exposure event. If parents are to be supported in navigating the testing process, ensuring that they understand the incubation period and also the difficulty of identifying an exposure event may be useful.

Our findings indicate that parents are still unsure about who is eligible for testing. Before NHSTT was launched widely to the public on May 28 2020, the UK Government prioritised testing for people in high-risk settings [17]. As testing capacity increased, the list of eligible people for testing increased. Changes have continued, particularly with the launch of lateral

flow tests in the UK for asymptomatic testing. Clarifying eligibility is likely to still be important.

Parents were largely supportive of getting the family tested and self-isolating when needed. However, previous research found that only 42.5% of people fully adhered to self-isolation, with adherence being lower in parents [3,4]. Our findings found family resources were a prominent factor determining the ability to adhere to self-isolation. A minority of parents tried to keep their home stocked with essentials, but the unpredictability of the situation made this an unreliable solution. Some parents appeared to adopt a partial approach to adherence by continuing essential errands whilst waiting for test results, something which might generate additional secondary cases [18]. Parents suggested they would speak to their friends and family about behaviours they felt were non-adherent to self-isolation, indicating adherence is considered a social norm [19]. Nevertheless, testing was impacted by stigma. Parents' perceptions of how others would react prevented them from informing others and some from getting a test, supporting research that found people were less likely to seek a COVID-19 test if they anticipated feeling stigmatised [20].

Parents worried about the impact of self-isolating on their job and their children's education, reducing intended adherence, especially for households with multiple self-isolating experiences. Multiple bouts of self-isolation may reduce parental intention to seek a test because worry about the cumulative impacts of self-isolation on their job and on their children's education may increase at each period of self-isolation.

In terms of implications, our findings suggest several steps could be taken to encourage symptomatic testing and improve adherence to self-isolation among parents and their children. First, misunderstandings existed regarding testing, particularly as to the eligibility and the symptoms of COVID-19 needed for parents to request a test. Regardless of the case definition of the COVID-19 symptoms used, our findings reinforce the importance of clarity for the public around what symptoms to watch for. Colleagues in communication teams or who engage with the media should cease using the vague word "symptoms" and replace it with a description of the exact symptoms that people should watch for. Messages should also highlight the need to test for the symptoms, even if mild, and not to make assumptions about the cause before deciding whether to take a test. From March 2021 children and their household members were eligible to take two lateral flow tests (LFT) a week. Advising parents that if in any doubt, they could use an LFT is likely to result in greater uptake than requiring them to request a PCR home test kit or attend a testing site. However, whether the improvements in uptake outweigh the lower sensitivity of lateral flow tests compared to PCRs is unknown, and a PCR test will still be needed after a positive LFT. There is some suggestion that a confirmatory PCR is unnecessary given evidence that LFT tests are more accurate than first thought, therefore this may not be a requirement in the near future [21]. Second, the official online sources (e.g., Government and the NHS websites) were the primary source of information for parents; accurate information from trusted sources must appear as a top result in search engines. Using clear branding in communications and distributing them through various social platforms, media channels, organisations, and schools will facilitate information distribution to parents and their children. Third, parents felt anxious whilst waiting for test results, and this was also a period where parents were more likely to be non-adherent. Ensuring a rapid turnaround time for test results is essential to reduce non-adherence and negative impacts on wellbeing. Fourth, parents who are prepared for, and supported in, self-isolation are more likely to be able to adhere. Pro-active support from Government and their sector agencies is essential. Fifth, investigating a process with a high public profile, which underwent a series of rapid changes (e.g., increasing the capacity to carry out tests from 2,000 per day to 790,000 nine months later [22]) was challenging during the interviews. We needed to incorporate how a participant's

experience of using NHSTT might have differed depending on when they engaged with the testing process. The functions of NHSTT will continue until at least March 2022 (when it will be reviewed) and it would be beneficial for a further investigation into NHSTT now that the process has been in place for a longer period of time [23].

## Limitations

One limitation of our study is the possibility of selection bias. Participants who are motivated to participate in studies about the pandemic may have a particular view that they want to express, be particularly well-informed, or be particularly motivated to want to help scientists bring it to an end. Therefore, it is possible that interviewing disinterested parents would have revealed other themes. A second limitation is the potential for social desirability bias, again limiting the themes that we identified. However, the results included some parents reporting their non-adherence, suggesting we could elicit non socially desirable responses [24]. Third, not all parents had experienced symptoms in themselves or their children, requiring them to report on intentions rather than behaviours. Fourth, participants were all required to be proficient in English, and therefore the results may not be generalisable to other populations.

## Conclusion

This study furthers research into the factors that promote or inhibit COVID-19 symptom identification, requesting a COVID-19 test and self-isolation. Parents appeared to be well informed in using NHSTT and we did not identify any clear gaps in communications between parents and the NHS. However, to promote engagement with test, trace, and isolate systems, trusted sources need to reflect key information in communications, such as reiterating the symptoms of COVID-19 and keeping messaging consistent. Information should emphasise that a test should be sought as soon as possible after a symptom of COVID-19 has been identified, regardless of the suspected cause. Non-adherence to self-isolating whilst waiting for test results appears somewhat socially acceptable. Highlighting that it is vital to self-isolate during this period may reduce transmission. Parents still face difficulties balancing disruptions to education and their job and preventing the onward transmission of infection. Providing families with access to resources they need, such as deliveries of groceries and necessary medicines, may facilitate self-isolation, particularly for families who have had multiple experiences of self-isolation.

## Supporting information

**S1 Text. Interview outline.** The list of questions used by the interviewer to guide the interview.
(DOCX)

## Author Contributions

**Conceptualization:** Lisa Woodland, Fiona Mowbray, Louise E. Smith, Rebecca K. Webster, Richard Amlôt, G James Rubin.

**Data curation:** Lisa Woodland.

**Formal analysis:** Lisa Woodland.

**Funding acquisition:** Rebecca K. Webster, Richard Amlôt, G James Rubin.

**Investigation:** Lisa Woodland, Fiona Mowbray.

**Methodology:** Lisa Woodland, Fiona Mowbray, Louise E. Smith, Rebecca K. Webster, Richard Amlôt, G James Rubin.

**Project administration:** Lisa Woodland, Fiona Mowbray.

**Supervision:** Rebecca K. Webster, Richard Amlôt, G James Rubin.

**Visualization:** Lisa Woodland.

**Writing – original draft:** Lisa Woodland.

**Writing – review & editing:** Fiona Mowbray, Louise E. Smith, Rebecca K. Webster, Richard Amlôt, G James Rubin.

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
