## [Decision Letter · Decision Letter 0]

29 Nov 2021

PONE-D-21-35530What influences whether parents recognise COVID-19 symptoms, request a test and self-isolate: A qualitative study.PLOS ONE

Dear Dr. Woodland,

Thank you for submitting your manuscript to PLOS ONE. After careful consideration, we feel that it has merit but does not fully meet PLOS ONE’s publication criteria as it currently stands. Therefore, we invite you to submit a revised version of the manuscript that addresses the points raised during the review process.

We look forward to receiving your revised manuscript.

Kind regards,

Sanjay Kumar Singh Patel, Ph.D.

Academic Editor

PLOS ONE

Journal Requirements:

https://journals.plos.org/plosone/s/file?

GJR, RA and LS participate in the UK’s Scientific Advisory Group for Emergencies, or its subgroups. RA is an employee of the UK Health Security Agency.  

We note that you received funding from a commercial source:  UK’s Scientific Advisory Group for Emergencies

Within this Competing Interests Statement, please confirm that this does not alter your adherence to all PLOS ONE policies on sharing data and materials by including the following statement: ""This does not alter our adherence to PLOS ONE policies on sharing data and materials.” (as detailed online in our guide for authors http://journals.plos.org/plosone/s/competing-interests).  If there are restrictions on sharing of data and/or materials, please state these. Please note that we cannot proceed with consideration of your article until this information has been declared. 

Reviewers' comments:

Reviewer's Responses to Questions

**Comments to the Author**

1. Is the manuscript technically sound, and do the data support the conclusions?

Reviewer #1: Yes

Reviewer #2: Yes

2. Has the statistical analysis been performed appropriately and rigorously? 

Reviewer #1: Yes

Reviewer #2: Yes

3. Have the authors made all data underlying the findings in their manuscript fully available?

Reviewer #1: Yes

Reviewer #2: Yes

4. Is the manuscript presented in an intelligible fashion and written in standard English?

Reviewer #1: Yes

Reviewer #2: Yes

5. Review Comments to the Author

Reviewer #1: The research article entitled, " What influences whether parents recognise COVID-19 symptoms, request a test and self-isolate: A qualitative study" by Woodland et al., assessed the factors linked with the use of England’s NHS Test and Trace service among parents of school-aged children by one-to-one telephone interviews with parents (n = 18) of school-aged (4 to 18 years) children living in England between 30 November to 11 December 2020. Authors found three themes and eight sub-themes in their analysis and identified several barriers to parents using NHS Test and Trace as needed. Finally, authors have concluded that Information about the eligibility of testing (main symptoms of COVID-19 and the age of eligibility) needs to be more precise and resources provided to enable families to adhere to self-isolation if the efficiency of test, trace and isolate systems is to be optimized. Altogether this is an important and timely research article, this reviewer has certain suggestions that would help produce a more comprehensive overview of the topic:

Comments:

1. Did authors inquire about any depression during their study period in participants? This will be noteworthy to know and authors can include this data to their study.

2. At least one supplementary Figure as illustration may be afforded as to highlight the summary or prospect of this study.

3. Author may explain how to adjust the limitation of selection bias in their study.

4. The abbreviations should be cross validated in the manuscript (First define them fully followed by abbreviation) and one paragraph can be added for abbreviations.

5. This study was done between 30 November to 11 December 2020 (very starting period of Covid19), did authors find any communication gap between NHS and the parents?

Reviewer #2: In this paper entitled "What influences whether parents recognize COVID-19 symptoms, request a test and self-isolate: A qualitative study.", the authors investigated the factors associated with the use of England NHS test and trace service among parents of school-aged children.The results identify several barriers to parents and the importance of information to adhere to self-isolation. The manuscript is good and well carried out. The manuscript addresses a research topic of great interest; however, it requires particular suggestions that would improve the manuscript:

Minor comments:

1) The study investigated parents (n=18), whereas Hodson, woodland et al. 2021 has interviewed 30 parents in England. Could the authors explain how his results are significant with a small sample size in the manuscript?

2) There are formatting errors in Tables 2 and 3.

3) The importance of the study may be specifically highlighted in the introduction.

4) The author may provide a paragraph regarding challenges or prospects of study in the manuscript.

5) Correct reference error in the manuscript.

6) A figure could be provided to summarise the results for better understanding.

6a) Although the number of parents is small. Still, the author can provide information about the number of patients, subtheme, description, frequency, and ratio in the tabular/graphical form in the result section.

---

## [Author Response · Author response to Decision Letter 0]

7 Jan 2022

Reviewer #1: The research article entitled, " What influences whether parents recognise COVID-19 symptoms, request a test and self-isolate: A qualitative study" by Woodland et al., assessed the factors linked with the use of England’s NHS Test and Trace service among parents of school-aged children by one-to-one telephone interviews with parents (n = 18) of school-aged (4 to 18 years) children living in England between 30 November to 11 December 2020. Authors found three themes and eight sub-themes in their analysis and identified several barriers to parents using NHS Test and Trace as needed. Finally, authors have concluded that Information about the eligibility of testing (main symptoms of COVID-19 and the age of eligibility) needs to be more precise and resources provided to enable families to adhere to self-isolation if the efficiency of test, trace and isolate systems is to be optimized. Altogether this is an important and timely research article, this reviewer has certain suggestions that would help produce a more comprehensive overview of the topic:

Comments:

1. Did authors inquire about any depression during their study period in participants? This will be noteworthy to know and authors can include this data to their study.

Response: We did not directly ask participants about depression or any other illnesses in our interviews. We agree that it may have been useful to investigate this in the study. However, mental illness was not the focus of the study and we only had scope to focus on questions relating to how parents used NHS Test and Trace. We purposively asked open-ended questions and parents were encouraged to talk about their own experiences of using the system, including how their mental or physical health may have impacted how they were using the system and we have reported on these findings. Furthermore, we describe the questions that we ask in the Method section and the interview guide is provided as supporting information. We feel that this is sufficient for the reader to understand what was discussed in the interviews. 

2. At least one supplementary Figure as illustration may be afforded as to highlight the summary or prospect of this study.

Response: We have added a table summarising the study findings. 

3. Author may explain how to adjust the limitation of selection bias in their study.

Selection bias is an inherent part of any qualitative study – we can only interview people who are willing to be interviewed. We have now expanded the limitations section of the study to ensure that readers are aware of this. 

4. The abbreviations should be cross validated in the manuscript (First define them fully followed by abbreviation) and one paragraph can be added for abbreviations.

Response: We have checked that all the abbreviations we use have been used as you have suggested and added a paragraph listing all the abbreviations we have used.

5. This study was done between 30 November to 11 December 2020 (very starting period of Covid19), did authors find any communication gap between NHS and the parents?

Response: We did not find a clear gap in communication between the NHS and parents. However, we did identify areas where parents had struggled to implement the NHS guidance in practice. For example, where we describe in the discussion (line 432 to 438) how parents needed prompting before they were able to recall the NHS main symptoms of COVID-19 and surrounding the eligibility of testing (line 465 to 470). We have added “parents appeared to be well informed about using NHSTT and we did not identify any clear gaps in communications between parents and the NHS” in the conclusion to distinguish between gaps in communications and misunderstandings.

Reviewer #2: In this paper entitled "What influences whether parents recognize COVID-19 symptoms, request a test and self-isolate: A qualitative study.", the authors investigated the factors associated with the use of England NHS test and trace service among parents of school-aged children. The results identify several barriers to parents and the importance of information to adhere to self-isolation. The manuscript is good and well carried out. The manuscript addresses a research topic of great interest; however, it requires particular suggestions that would improve the manuscript:

Comments:

1) The study investigated parents (n=18), whereas Hodson, Woodland et al. 2021 has interviewed 30 parents in England. Could the authors explain how his results are significant with a small sample size in the manuscript?

Response: We agree that when using qualitative methodology there is no clear sample size that confirms the findings are significant and ensuring data saturation has been reached. We tried to mitigate this issue as much as possible by using Fugard and Potts (2015) ‘supporting thinking on sample sizes for thematic analyses: a quantitative tool’ to provide a high likelihood of reaching data saturation. We met the suggested number of participants; therefore, we can assume that the same themes would be identified if we had interviewed more participants. Hodson, Woodland et al. 2021 had different aims, objectives, and methodological parameters resulting in a larger sample size being used and therefore we cannot use this as a guide for the sample size we used in this study. 

2) There are formatting errors in Tables 2 and 3.

Response: Thank you for identifying the error we have reformatted our tables to ensure they are displayed accurately. 

3) The importance of the study may be specifically highlighted in the introduction.

Response: To make the importance of our study clearer, we have added the sentence “these difficulties highlight the importance of identifying areas within the NHSTT process to increase parents’ engagement,” following the paragraph in the introduction about how COVID-19 has negatively impacted children’s health and education (line 90 to 92). 

4) The author may provide a paragraph regarding challenges or prospects of study in the manuscript.

Response: Within our paragraph on study implications in the discussion we have added the below section describing future areas of study (line 491+): 

“Fifth, investigating a process with a high public profile, which underwent a series of rapid changes (e.g., increasing the capacity to carry out tests from 2,000 per day to 790,000 nine months later [22]) was challenging during the interviews. We needed to incorporate how a participant’s experience of using NHSTT might have differed depending on when they engaged with the testing process. The functions of NHSTT will continue until at least March 2022 (when it will be reviewed) and it would be beneficial for a further investigation into NHSTT now that the process has been in place for a longer period of time [23].”

5) Correct reference error in the manuscript.

Response: Thank you for identifying the error in our references. We have resolved this error and re-checked our in-text citations and references to ensure they are accurate. 

6) A figure could be provided to summarise the results for better understanding.

Response: We have included a table that summarises the results. 

6a) Although the number of parents is small. Still, the author can provide information about the number of patients, subtheme, description, frequency, and ratio in the tabular/graphical form in the result section.

Response: Given that the data are inherently qualitative in nature, overemphasising quantitative aspects would be misleading. As noted, the sample is not a large representative sample and our aim is to articulate the breadth of issues that are relevant, and not to quantify number of parents who experience each issue. We have, however, now provided a table of the key themes which we hope goes some way towards clarifying the results.

---

## [Editor Report · Decision Letter 1]

21 Jan 2022

What influences whether parents recognise COVID-19 symptoms, request a test and self-isolate: A qualitative study.

PONE-D-21-35530R1

Dear Dr. Woodland,

We’re pleased to inform you that your manuscript has been judged scientifically suitable for publication and will be formally accepted for publication once it meets all outstanding technical requirements.

Kind regards,

Sanjay Kumar Singh Patel, Ph.D.

Academic Editor

PLOS ONE

---

## [Editor Report · Acceptance letter]

28 Jan 2022

PONE-D-21-35530R1 

What influences whether parents recognise COVID-19 symptoms, request a test and self-isolate: A qualitative study. 

Dear Dr. Woodland:

I'm pleased to inform you that your manuscript has been deemed suitable for publication in PLOS ONE. Congratulations! Your manuscript is now with our production department. 

Kind regards, 

on behalf of

Dr. Sanjay Kumar Singh Patel 

Academic Editor

PLOS ONE